# Macromolecular Crowding Increases the Affinity of the PHD of ING4 for the Histone H3K4me3 Mark

**DOI:** 10.3390/biom10020234

**Published:** 2020-02-04

**Authors:** Alicia Palacios, Francisco J Blanco

**Affiliations:** 1CIC bioGUNE, Bizkaia Science and Technology Park, bld 800, 48160 Derio, Bizkaia, Spain; alicia.palacios@unir.net; 2IKERBASQUE, Basque Foundation for Science, Maria Diaz de Haro 3, 6 solairua, 48013 Bilbao, Bizkaia, Spain

**Keywords:** macromolecular crowding, PHD, ING4, histone H3, H3K4me3, NMR, protein–protein interaction

## Abstract

The five members of the family of tumor suppressors ING contain a Plant Homeodomain (PHD) that specifically recognizes histone H3 trimethylated at lysine 4 (H3K4me3) with an affinity in the low micromolar range. Here, we use NMR to show that in the presence of 15% Ficoll 70, an inert macromolecular crowding agent, the mode of binding does not change but the affinity increases by one order of magnitude. The affinity increases also for unmethylated histone H3 tail, but the difference with H3K4me3 is larger in the presence of Ficoll. These results indicate that in the cellular milieu, the affinity of the ING proteins for their chromatin target is larger than previously thought.

## 1. Introduction

The histone code hypothesis states that the post-translational modifications of histones function to direct specific and distinct nuclear processes [1,2]. Histones are post-translationally modified by specific enzymes that “write” the code by adding or removing different chemical groups (methyl and acetyl being the most abundant) on specific histone amino acids, most frequently in their disordered N-terminal and C-terminal tails [3,4]. The histone code is “read” by certain conserved domains that selectively bind the modifications. These domains recruit chromatin remodeling complexes driving gene transcription regulation and other chromatin related processes [1,4]. One of these domains is the Plant HomeoDomain (PHD), which specifically recognizes histone H3 methylated at lysine 4, a modification involved in transcription activation [5,6].

The tumor suppressor family Inhibitors of Growth [7] is involved in chromatin remodeling [8], and its members contain a PHD motif at their C-terminal ends that specifically bind histone H3 trimethylated at lysine 4 (H3K4me3) [9]. ING4 is a member of this family that forms dimers through its N-terminal domain [10,11] and binds DNA through its disordered central region [12]. We have previously determined the molecular basis of H3K4me3 recognition by the PHD of ING4, revealing that methylated versus nonmethylated histone tail discrimination by ING4 is predominantly of entropic nature likely due to solvation effects [13,14]. Still the affinity for H3K4me3 is not as high as other physiological protein–protein interactions. These binding studies have been performed in solutions containing low protein concentration, but, inside the cells, the macromolecular crowding environment is an important factor in protein interactions [15,16].

## 2. Materials and Methods

The preparation of the ING4-PHD, the stocks of peptides, the titration procedure, and the analysis of the chemical shift perturbations measured on peptide addition were done as previously described [13]. The histone H3 peptide used corresponds to residues 1–15, with an extra Tyr residue at the C-terminus (ARTKQTARKSTGGKAY) for accurate peptide concentration measurement by UV-light absorbance. The extinction coefficients of both the PHD and the peptide were calculated from the amino acid composition using the ProtParam web server (https://web.expasy.org/protparam/). A stock of Ficoll 70 (Sigma) at 20 % (*w*/*v*) was prepared in 20 mM sodium phosphate pH 6.5, 50 mM NaCl, and was used to prepare the different NMR samples. These contained 50 or 10 µM uniformly -^15^N enriched PHD in 20 mM sodium phosphate pH 6.5, 50 mM NaCl, 1 mM perdeuterated dithiotreitol, 5% (*v*/*v*) ^2^H_2_O, and 0.01% (*w*/*v*) NaN_3_, plus different concentrations of Ficoll 70. The stability of the solutions of ING4-PHD/H3K4meX peptides (at 1:4 molar ratios) in the presence of 15 % Ficoll 70 was evaluated under the same conditions used for the NMR measurements by monitoring the absorbance of the sample at 280 nm every 8 h over a period of several days. The absorbance was unchanged during the first 40 h, with a gradual reduction in the absorbance afterwards, and the appearance of visible precipitated material at longer times. Thus, the NMR titrations were completed in less than 24 h. NMR spectra were recorded at 25 °C on a BRUKER Avance III 800 spectrometer equipped with a TCI cryogenically cooled probe. At a concentration of 50 µM PHD, ^1^H-^15^N Heteronuclear Single Quantum Coherence (HSQC) spectra could be recorded in about 2 h with sufficient quality to measure the chemicals shift perturbations on peptide addition. At 10 µM PHD concentration, HSQC were recorded with only 16 t_1_ points and a ^15^N spectral width of 10 ppm. This caused a number of signals to appear folded in the indirect dimension, but the carrier was positioned so that the backbone amide signal of W237 was not folded and did not overlap with folded signals over the entire titration. In this way, the full titration of ING4-PHD with the histone peptide could be measured in less than 24 h. These measurements could be fitted to a single site binding curve with an error in the fitting that allowed for the observation of significant differences among the peptides and with the concentration of Ficoll 70.

## 3. Results

To explore the molecular recognition properties of the PHD of ING4 in a macromolecular crowded environment we used Ficoll 70, a neutral and highly branched polar polysaccharide frequently used at 15% concentration to mimic macromolecular crowding studies [17,18]. It behaves like a semirigid sphere, is chemically inert, and does not interact with proteins [19].

We first recorded NMR experiments of 50 µM ING4-PHD with and without H3K4me3 peptide (1:4 protein:peptide molar ratio) in the same buffer with or without 15% Ficoll 70. The pattern of chemical sift perturbations in the NMR signals of the PHD caused by the peptide is the same in the presence or in the absence of Ficoll 70 (Figure 1). Only a few residues show Ficoll induced CSP slightly above the experimental error (Appendix A), most of them around residue Glu220, which showed a certain degree of conformational exchange in free PHD [14]. This result indicates that the structure of the complex is not affected by the presence of 15% Ficoll 70, and therefore that it behaves as an inert molecule that does not perturb the mode of binding of the PHD to the H3K4me3 peptide.

Next, we measured the affinity of the binding in presence of Ficoll, by means of a NMR titration of the PHD with H3K4me3 peptide but in presence of 15% Ficoll (Figure 2). For binding analysis, we used the backbone amide signal of amino acid W237 because it experiences a large CSP upon peptide binding and is well isolated in a region of the spectrum with few other signals. The chemical shift perturbation observed in the ^1^H-^15^N HSQC spectra of the ING4 PHD for W237 with the addition of increasing amounts of peptide can be represented obtaining a saturation curve, but the data cannot be fitted to a 1:1 equilibrium because the affinity increased by an order of magnitude (see below) and at 50 µM PHD concentration the curvature is too low for a good fitting. This makes it necessary to reduce the protein concentration of the experiment.

On the other hand, the presence of 15% Ficoll 70 lowered the signal to noise ratio of the NMR experiment as signals got broader due to the increased viscosity of sample. So, we had to decrease the protein concentration of the experiment but not so much as to lose the signals in the noise. After several trials, we found that 10 µM PHD concentration was a good compromise between a reasonable signal to noise ratio and a good fit of the CSP data. In order to reduce the total acquisition time used for each NMR experiment along the titration, we reduced the number of points in the indirect dimension, and to avoid an excessive loss of resolution we accordingly reduced the indirect dimension sweep width, taking care that the amide signal of W237 was not perturbed by other signals folded in the indirect dimension. In these conditions, we could record ^1^H-^15^N HSQC experiments in 2 h and complete the titration in less than 1 day.

Once we got the optimum conditions, we did the titration by NMR on a sample of 10 µM PHD of ING4 in the presence of increasing amounts of H3K4me3 peptide (Figure 3). The result was a K_D_ = 0.4 ± 0.3 µM, an increase in the affinity by one order of magnitude compared to the binding affinity in absence of Ficoll (K_D_ = 3.9 ± 0.7 µM). This is consistent with the increase in affinity measured by fluorescence, with a change in K_D_ from 8 ± 2 µM [9] to 0.4 ± 1.1 µM (Appendix A).

Titrations at different percentages of Ficoll (0, 5, and 10 %) were done by NMR in a same manner as the titration at 15% Ficoll. The results show that the binding affinity increases exponentially with the concentration of Ficoll (inset in Figure 3 showing the linear dependence of log K_D_); thus, large changes in the affinity are observed when the concentration of macromolecules occupies high percentages of the total volume.

We also examined the increase in affinity for histone H3 with different methylation states of residue K4 (H3K4me0/1/2/3). Each curve in Figure 4 corresponds to the chemical shift perturbation in the PHD W237 NMR signal upon the addition of each peptide, and gives us the histone H3 binding constant. We observe that the K_D_ for H3K4me0 (100 ± 10 µM) decreases by a factor of 4 with respect to the value in the absence of Ficoll (K_D_ = 370 ± 20 µM [14]). Thus, the PHD discrimination between methylated and nonmethylated state of histone H3 is higher in presence of 15% Ficoll 70.

## 4. Discussion

The binding of the PHD domain of ING proteins to the histone H3K4me3 mark is in the low micromolar range as measured with the isolated domains in solution [9]. At least ING4 and ING5 full-length proteins have been shown to form dimers. The conserved N-terminal region of ING4 forms an antiparallel coiled-coil dimeric structure, with the two PHD pointing to opposite directions but tethered to the coiled-coil by a long disordered region. Therefore, ING4 is a bivalent reader of the histone H3K4me3 [10]. The full length ING4 binds H3K4me3 in the same way and with a three-fold higher affinity than the isolated PHD finger [10]. The homologous ING5 dimer also binds H3K4me3 in the same way and with a three-fold higher affinity than its isolated PHD finger [20]. We interpret the slightly higher affinity of the full-length proteins to the presence of two identical PHD fingers in the ING dimer. Still, the affinity for H3K4me3 is not as high as other physiological protein–protein interactions. These binding studies have been performed in solutions containing low protein concentrations, but, inside the cells, the macromolecular crowding environment might be an important factor in ING proteins’ interactions with histone H3 [16,21]. In fact, the affinity of some protein–peptide interactions have been measured by FRET-based techniques to be an order of magnitude larger in the intracellular medium than measured on the isolated protein in dilute buffered solutions [22]. We have observed the same affinity increase in 15% Ficoll 70 for the interaction between the ING4 PHD finger and H3K4me3, which strongly indicates that inside the cell the interaction is of higher affinity than previously thought. The actual ligand of ING4 is the nucleosome, consisting of a histone octamer with two copies of histone H3 and a 147 bp-long DNA duplex. Therefore, the overall affinity will even higher because of two cooperative effects: the micromolar affinity binding of the central region of ING4 to DNA and the presence of two histone H3 N-terminal tails in each nucleosome [23]. ING4 does not work alone but as member of the histone acetyl transferase HBO1 complex, which includes the JADE proteins with PHD fingers that also bind to histone H3 N-terminal tail [24]. Therefore, the affinity of the HBO1 complex for the chromatin could be very high.

## 5. Conclusions

We find a strong effect of macromolecular crowding in the affinity of histone H3K4me3 recognition by the PHD of ING4. This fact is a further demonstration that the measurements in vitro are only able to give us a reasonable approximation of the real situation that takes place inside the cells.

## Figures and Tables

**Figure 1 biomolecules-10-00234-f001:**
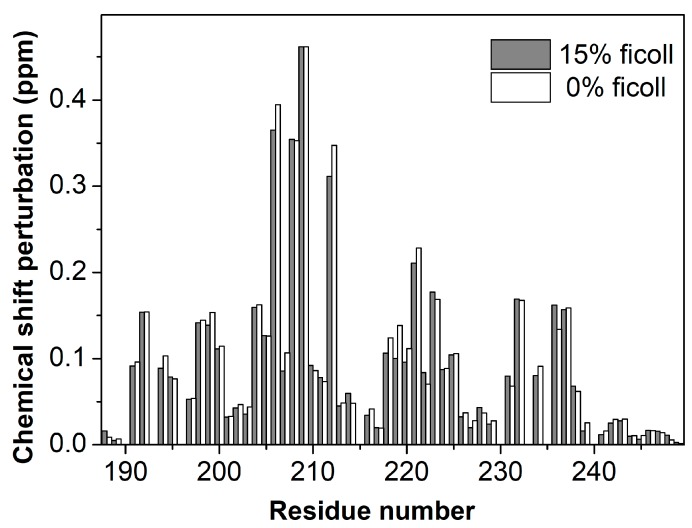
Comparison of Chemical Shift Perturbation (CSP) caused by histone H3 trimethylated at lysine 4 (H3K4me3) peptide binding to ING4- Plant Homeodomain (PHD). The bar plot shows the CSP observed for each residue in ^1^H-^15^N HSQC spectra of the PHD in the presence of a four-fold molar excess of H3K4me3 with 15% (grey bars) or 0% (white bars) Ficoll 70. The estimated experimental error is 0.008 ppm.

**Figure 2 biomolecules-10-00234-f002:**
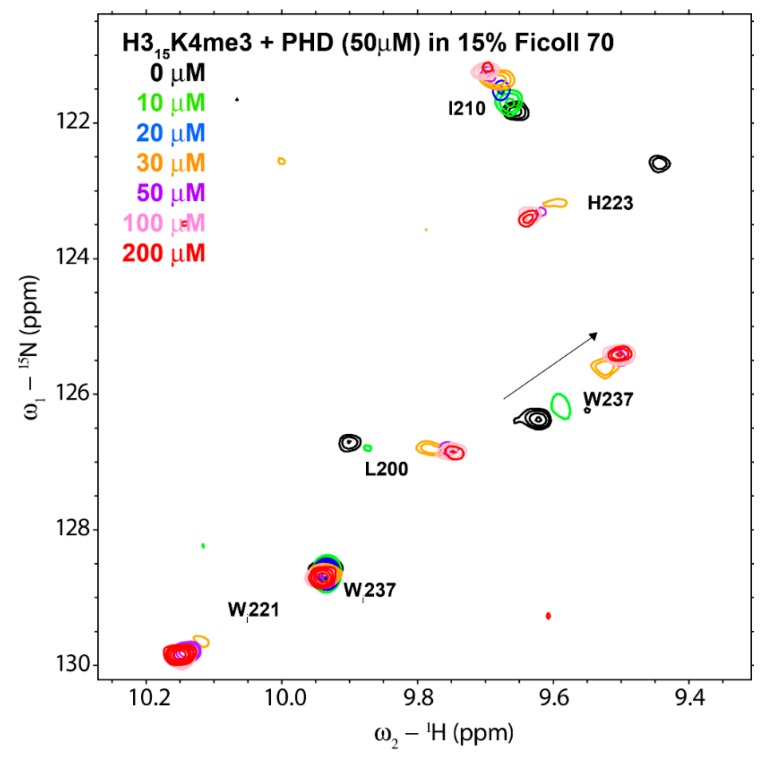
Superimposition of a region of ^1^H-^15^N HSQC spectra of ING4-PHD after addition of different amounts of H3K4me3 peptide, indicated with different colors. The labels adjacent to the signals indicate the PHD residue (“W_i_” stands for the tryptophan indole NH signals). Conditions: 50 µM PHD in 20 mM sodium phosphate pH 6.5, 50 mM NaCl, 1 mM perdeuterated dithiothreitol, 15% Ficoll 70, 5% ^2^H_2_O, and 0.01% NaN_3_, at 25 °C.

**Figure 3 biomolecules-10-00234-f003:**
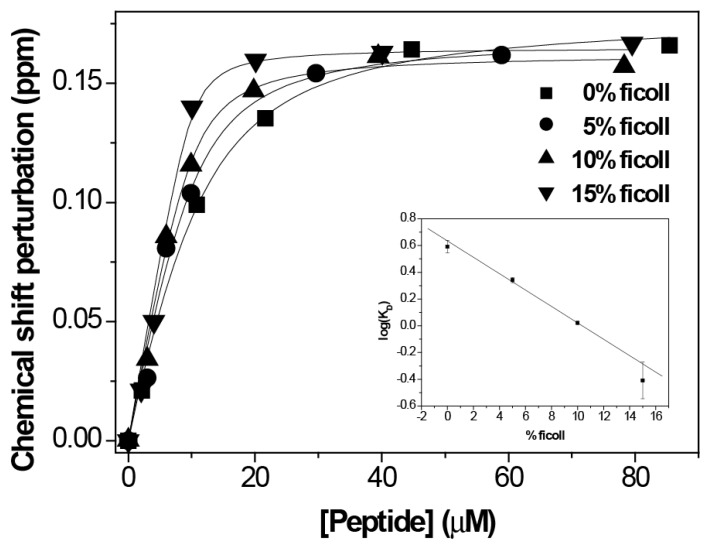
NMR analysis of the binding of PHD to histone H3K4me3 in the presence of different amounts of Ficoll 70. The CSP of the W237 amide resonance of ^1^H-^15^N-HSQC spectra of ING4 PHD is represented against the peptide concentration for each titration. The continuous lines are the fittings to the binding model. The height of the symbols corresponds with the experimental error of the measurements. Inset, log K_D_ versus % of Ficoll 70.

**Figure 4 biomolecules-10-00234-f004:**
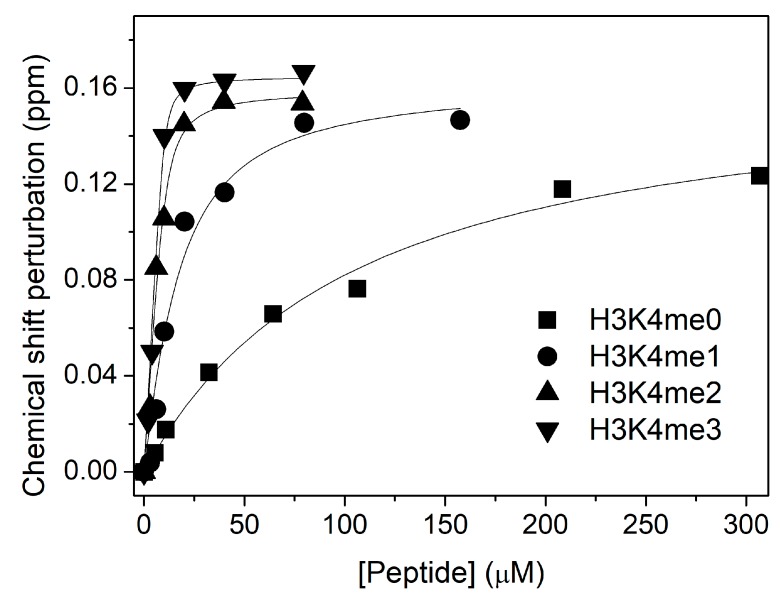
NMR analysis of the binding of ING4 PHD to H3K4me0/1/2/3 peptides in presence of 15% Ficoll 70. The CSP of W237 amide resonance of the PHD in ^1^H-^15^N-HSQC spectra is represented as a function of peptide concentration for each titration. The continuous lines are the fittings to the binding model, and the height of the symbols corresponds with the experimental error. The measured K_D_ values are 100 ± 10, 12 ± 4, 1.4 ± 0.6, and 0.4 ± 0.3 µM, respectively.

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
