# Peer review of "Macromolecular Crowding Increases the Affinity of the PHD of ING4 for the Histone H3K4me3 Mark"

_biomolecules, 2020, doi:10.3390/biom10020234_

Round 1

Reviewer 1 Report

The authors present a brief study investigating the affect of molecular crowding on the biding affinity of the ING4 PHD finger proteins.  The idea is conceptually interesting and they use Ficoll 70 to approximate the cellular environment, and test the binding and protein folding by NMR techniques.  They conclude that the binding mode is unchanged, but that molecular crowding increases the binding affinity for the ING4 PHD finger for the histone H3K4me3 and unmodified histone H3 ligands, thus within the cell the binding may be stronger than what has been previously reported by other in vitro studies. The data do appear to support the conclusions made, but further analysis and re-writing are required before this manuscript is ready for publication.  My suggestions are as follows:

Line 26, correct N-t to N-terminal

Have they tested to see the binding affinity of the histone peptides with and without the Y residue on the end of the histone H3 peptide to make sure it does not interfere with binding? They should be able to calculate the peptide concentration from the MW and molar mass used to make the stock rather than via UV.

What is U-15N-PHD?, page 2 line 52?

Line 77-78 they mention the pattern of CSPs in the NMR signals caused by the peptide is, within error, the same in the presence or absence of Ficoll 70. However, they do not have any error bars in Figure 1. Also, I would recommend that they show the CSP pattern of the protein with no ligand bound with and without Ficoll 70. Then, use the current Figure 1 to show how the CSPs are changed in the presence of Ficoll 70

There is no mention as to why they focus on W237 for the binding analysis. Also, the binding of one residue may not represent the affinity for the entire domain. For overall binding affinity it may make more sense to analyze the Kd for 4-5 residues and then use the average to calculate the binding affinity to the histone peptide.

Line 104 should read “In order to reduce the total acquisition…

Line 137-138, there appear to be some author comments here that should be removed.

Line 137, the conclusion should be that the discrimination between methylated and non-methlyated histone ligands is reduced in the presence of Ficoll 70 since with Ficoll the Kd for H3 unmod is 100 uM but without it is 370 uM, thus adding Ficoll 70 improves the binding interaction for unmodified peptides. The difference between 0.4 um and 100 uM is smaller than the difference between 4 uM and 370 uM.

It would be nice to include a table of the binding affinities reported to compare them more readily.

Typo line 158- actual ligand o ING4

Reviewer 2 Report

In this work the authors study the effect of molecular crowding on the affinity between a histone H3 peptide and the PHD finger domain of ING4. By NMR titrations they show that in the presence of 15% Ficoll the binding mode does not change but the affinity increases by one order of magnitude.
The results are interesting, as they suggest that inside the cells crowding conditions might increase interactions between histones and their readers.

The NMR experiments are sound.
My major criticisism consists in the fact that affinities are measured only with one method (NMR). As the main message of the manuscript is that crowding agents can change affinities by one order of magnitude, the authors should support their finding by orthogonal methods (e.g. ITC or fluorescence, whereby ITC could also help in making some considerations on the effect of crowding on entropy changes).
To generalize their finding the authors could also think to repeat one titration in the presence of another crowding agent (PEG, Dextran?)

Round 2

Reviewer 1 Report

I believe the manuscript has been improved enough to warrant publication.  However, I do agree with the other reviewer that an additional method to confirm the binding trends observed by NMR would strengthen the conclusion.  The authors could easily do it by Trp fluorescence as they mention they used this method previously to measure the binding affinities. Using this method would also specifically focus on the W237 residue highlighted in the current manuscript.

All my best,

Karen

Author Response

We have confirmed by fluorescence measurements the increase in affinity of the PHD finger of ING4 for histone H3K4me3 in the presence of 15% Ficoll 70 (new supplementary figure 2).

Reviewer 2 Report

I am convinced that another technique (considering also the low Kd, which is not ideal for NMR measurements) would have strengthen the results and supported the interesting conclusions.

Author Response

(The authors gave the same response as above.)
